# Proposing a Hybrid Authoring Interface for AR-Supported Human-Robot Collaboration

Rasmus Lunding
Aarhus University, Denmark

Sebastian Hubenschmid
University of Konstanz, Germany

Tiare Feuchtner
University of Konstanz, Germany
Aarhus University, Denmark

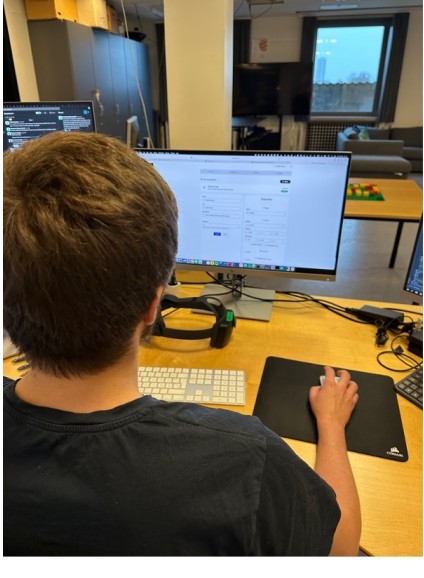 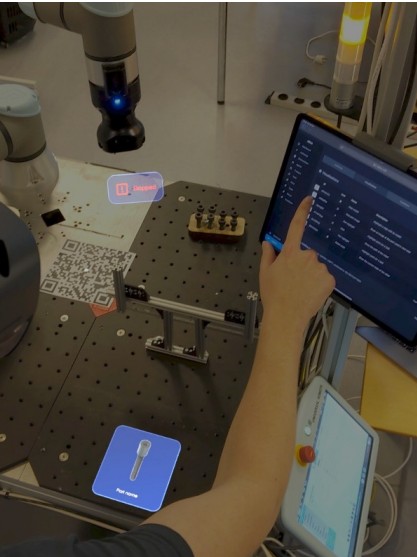 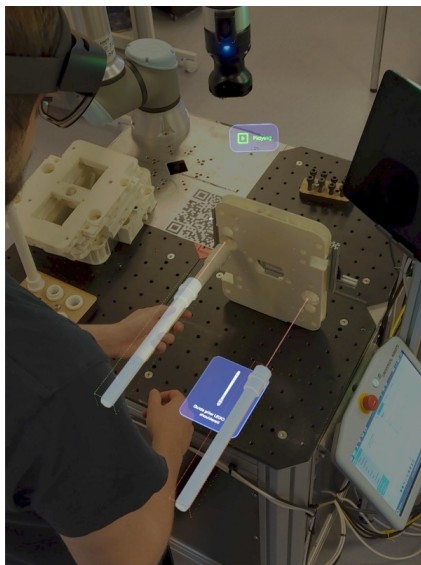

Figure 1: We support authoring of augmented reality content for human-robot collaboration scenarios through a hybrid user interface: (1) effective content creation can be performed through traditional interfaces (e.g., left: desktop computer, center: tablet); (2) inspection and adjustment of live virtual content can be done directly in the workspace using a head-mounted display (center); (3) the assembly task with the authored guidance can directly be executed in the application (right).

## ABSTRACT

We propose the use of a hybrid user interface for authoring augmented reality (AR) guidance in human-robot collaboration scenarios. When designing AR applications, e.g., for the HoloLens, the visual appearance of virtual content in the editor (e.g., Unity) typically differs dramatically from the real experience. Further, app deployment can be long-winded and therefore detrimental to quick design iterations. To address this inefficient procedure, we propose an authoring tool based on a web-interface, whereby the appearance of the virtual content is immediately updated in AR on a head-mounted display. This hybrid authoring interface approach thereby facilitates the use of traditional input methods (e.g., desktop computer, mobile touch-screen device) for the completion of complex authoring tasks through the web-interface, while supporting in-situ inspection with a head-mounted display.

## CCS CONCEPTS

• **Human-centered computing → Mixed / augmented reality**;
• **Computer systems organization** → Robotics.

## KEYWORDS

Human-Robot Collaboration; Human-Robot Interaction; Augmented Reality; Authoring; Guidance; Intent; Hybrid User Interface

**ACM Reference Format:**
Rasmus Lunding, Sebastian Hubenschmid, and Tiare Feuchtner. 2024. Proposing a Hybrid Authoring Interface for AR-Supported Human-Robot Collaboration. In *Proceedings of VAM-HRI (VAM-HRI'24)*. ACM, New York, NY, USA, 6 pages.

## 1 INTRODUCTION

Recent research has demonstrated the benefits of collaborative robots (cobots) for assisting with assembly tasks [22, 23], which allow for a tight collaboration between human and robots without extensive safeguards [8]. To make full use of cobots, the operator must be made aware of its ongoing and planned procedures and

be able to communicate with it. Here, information presented in augmented reality (AR) environments can be beneficial to convey the robot's intent [24, 26], visualize safety information, and highlight the procedures or tasks which the user has to do [22]. In our experience, such immersive human-robot collaboration (HRC) environments are typically authored on a desktop system (e.g., Unity3D), making full use of familiar input technology such as mouse and keyboard. However, effectively designing AR content in the editor requires accurate simulation of the real-world environment in which it should be anchored. Even then, there is usually a notable discrepancy between the simulation of AR content in Unity and its situated visualization upon deployment on the AR head-mounted display (HMD). Further, the process of deploying to the HMD, testing and debugging, and again editing on the desktop system is often slow, tedious, and challenging. To address this, prior work has recently demonstrated the potential of using augmented reality for authoring the AR-experience *in-situ* [21].

But what are the benefits of an in-situ AR authoring tool, which may arguably be limited in terms of input capabilities (mid-air interaction vs. mouse and keyboard)? We see several advantages: (1) the user can immediately test out variations of e.g., visualizations and their appearance properties, without needing to recompile the entire application. Furthermore, (2) such an authoring tool may lower the barrier of entry, as it can allow novice users to design AR-supported systems without requiring extensive programming skills. However, our experience from developing an authoring system in AR [21] has shown that the design and configuration of an entire AR experience can be difficult in a pure AR environment: for example, effective text entry on a virtual keyboard presented in mid-air is not trivial [12], and manipulation of traditional interface elements (e.g., buttons and lists) can be strenuous and difficult due to imprecise mid-air interaction [7]. Recent works have therefore argued for the combination of multiple devices as complementary interfaces [9, 27] (e.g., AR HMDs and mobile touchscreen devices) to leverage the respective advantages of each technology.

Building on our prior RoboVisAR [21] project, we therefore propose a hybrid authoring tool that includes a desktop environment, tablet, and an AR HMD to collectively author and run the complete AR user experience. The system should allow users to design all AR content, while directly working together with a robot. Besides enabling the initial setup and in-situ authoring, our proposed system fluidly supports the execution phase, allowing users to instantaneously try out their authored workflows on one or more robots, which may but do not have to be cobots. In the following, we present our concept, describing the features of our prototype.

For this paper, we limit our scope to authoring and collaborative assembly, thereby excluding other key aspects of HRC assembly (e.g., robot programming, designing assembly instructions). However, we are confident that in future these steps can be integrated with our concept.

## 2 RELATED WORK

The following sections investigate prior approaches for *authoring AR experiences for human-robot interaction (HRI)* and review the concept of *hybrid user interfaces*.

### 2.1 AR Authoring for HRI

In the context of authoring for HRI, AR environments have primarily been explored to assist non-technical users with defining the robots' behavior. Some noteworthy examples of these authoring tools are: GhostAR [5], V.Ra [6], and KineticAR [11]. However, these are concerned with robot programming and aspects related to that, but not the authoring of the AR content that supports HRI itself.

We address this gap in research with our own prior work RoboVisAR [21], with which we presented an AR authoring tool that enables the user to create situated visualizations of robot data (e.g., status and movement path). We employed a timeline based approach, which also can be found in previous work [19, 20], where a recording is made by executing the robot program before the authoring process begins. AR content is then designed using the data of this recording, before deployment for live execution. The system runs on a Microsoft HoloLens 2, and all interaction and feedback happens through this immersive interface with mid-air interaction. In our view, this system has three key limitations: (1) it is limited to 'robot visualizations', thus the visualization of assembly instructions is not supported; (2) the system does not support input from the user to the system when running, thus only information from the robot/system to the operator is supported and not the other way around; and (3) challenges of mid-air interaction, in particular when manipulating 2D user interface elements, can deteriorate the user experience [21, 22]. We aim to address these limitations with the *hybrid authoring interface* we describe in this paper.

### 2.2 Hybrid User Interfaces

Hybrid user interfaces combine *"heterogeneous display and interaction device technologies"* [10], such as AR HMDs with smartphones, tablets, or desktop systems, for complementary use. The potential of AR in HRI makes this combination especially compelling, as commonly-used devices (e.g., tablets) can be seamlessly extended with superimposed content, without restricting their use (e.g., [16, 18, 25]). In addition, hybrid user interfaces have shown to provide better performance for two-dimensional input such as text entry [12] and navigation [4], likely due to high familiarity [3, 14] and the availability of haptic feedback [17]. In the context of switching between the authoring environment on a desktop computer and AR deployment for inspection of the content, recent work has explored the asynchronous use of hybrid user interfaces [15]: For example, Hubenschmid et al. [13] combined a familiar 2D desktop interface for visual analytics with immersive virtual reality, to allow traditional data visualization on a 2D screen, as well as in-situ in 3D. This allows users to flexibly switch to the appropriate interface (i.e., 2D ex-situ or 3D in-situ), as the tasks demand. With the current work, we propose to apply this hybrid authoring approach to the use case of HRI in manufacturing workflows.

## 3 AUTHORING AR CONTENT IN A HYBRID USER INTERFACE

With our work, we aim to support the authoring of AR content to facilitate HRC in the context of assembly procedures in manufacturing. For example, conveying the robot's intent by indicating its status (see Figure 1, right), highlighting the next component it will pick up, or visualizing its movement path, will prevent startling

the worker and enable them to adapt their own behavior, e.g., by getting out of the robot's way. However, when and how such information should be visualized is not trivial, and the design of such AR systems remains a tedious task.

The hybrid authoring tool we propose, involves the use of an optical-see-through AR HMD, a tablet, and a desktop computer. Each of these interfaces has specific affordances and limitations, which we aim to compensate through complementary use. By supporting mid-air interaction through hand tracking, the HMD leaves the user's hands available for manipulating physical objects in the workspace. However, this entails the drawback of mid-air interaction, e.g., poor ergonomics [2] and the lack of haptic feedback [7], in particular when interacting with 2D UI elements, such as menus. Hence, we propose the use of a tablet for all menu interaction, facilitating effective manipulation of common UI elements, including sliders, buttons, and text input, through well-established touchscreen interaction. Such a mobile device (e.g., tablet or phone) could also directly be used for displaying AR content. However, the non-stereoscopic display poses challenges for depth-perception, and the manual handling of this "peephole" can be tedious and may occupy the user's hands. Finally, we aim to support a desktop computer interface for more complex parts of the authoring procedure, as this remains the default system for the design and implementation of interactive AR systems. While, compared to the tablet and HMD, this interface obviously lacks the affordance of portability and in-situ visualization of AR content, the user can profit from superior input performance with mouse and keyboard, as well as greater processing power and better interoperability with other applications. By combining these complementary interfaces, users can benefit from the appropriate modalities, while also seamlessly switching between these devices to best suit their current task.

## 3.1 AR cues for Human-Robot Collaboration

Before describing the interfaces in more detail, we provide a short explanation of the design elements we envision. As communication is an important part of collaboration, the robot must be able to communicate with the operator and vice versa. We propose using AR as the main feedback channel, enabling the robot to communicate with the user, thus *visualizations* are an important component of the system. To respond, the operator communicates with the robot through so-called *actions*. Besides visualizations and actions, we adopt the idea from RoboVisAR [21] that *conditions* are used to trigger actions and control when visualizations are shown. All three elements have a set of *properties*, some of which can be edited by the user.

*Visualizations.* According to Suzuki et al., AR visualizations for HRI can be menus, points and locations, paths and trajectories, and areas and boundaries [26]. As the context is highly relevant for deciding which visualizations are most meaningful, a reasonable number of these should exist in any authoring tool, and the possibilities to adjust them should be plentiful.

We therefore intend to support a variety of visualizations related to the robot, the assembly task, and general purpose. Robot related visualizations could be: *movement path*, *robot silhouette* showing a preview of movement, *robot status*, and *sensor readings*

(e.g., force/torque). Assembly-related visualizations could be: step-by-step instructions, highlighting of parts (e.g. bolts) and tools, and textual instructions. Finally, general purpose visualizations could be different types of 3D objects, e.g. conveying safety zones or otherwise highlighting areas of interest in space, and user defined textual information.

*Actions.* Actions allow the user to communicate to the system, by defining commands that should be reacted upon. What events are relevant again highly depends on the task and the possibilities provided by the respective setup. For example, direct robot control might not be desirable due to safety concerns, but changing the speed of the robot, momentarily pausing the robot, or changing its current task might all be relevant actions. We intend to support at least three categories of actions: *robot related*, *assembly related*, and *general purpose actions*.

*Conditions.* Conditions are used to control when visualizations are shown and to trigger actions. Thus they play an important role in preventing visual clutter and ensuring that relevant information is shown. The following are examples of conditions we aim to support:

- *Proximity condition*: Triggers when two user-defined anchors are within a defined distance of each other (e.g., show a notification when the operator is within three meters of the workstation).
- *Robot assistance*: Triggers when the robot requires the operator to perform an action (e.g., triggered by the robot-program if a material dispenser is empty). This could then show a message/warning to the user and play a notification sound.
- Gaze+pinch: Some visualizations can be interacted with using gaze and pinch. For example, activating the "Done" button on a 2D UI panel could trigger a complete-task-action.

We intend to support conditions of the following categories: *spatial condition* (e.g., proximity, inside box and stationary), *robot condition* (e.g., robot state, assistance needed, sensor values like force/torque), *user condition* (e.g., skill-level, gaze), and *assembly condition* (e.g., task available).

*Properties.* Visualizations, events, and conditions all have various properties that might be editable by the user. An example is illustrated in Figure 2, where the anchor and an offset position relative to the anchor can be set. Changes to the value of a property should immediately be reflected across all interfaces, i.e., it must also be applied directly to the visualized AR content.

## 3.2 Hybrid Authoring Interface Components

We have designed two distinct graphical user interfaces to facilitate interaction with the hybrid authoring system: (1) a web interface for use at a desktop computer or on a touch-device, like a tablet; (2) an AR application running on an HMD.

*Web interface.* The web interface allows users to author all aspects of the AR experience and setup the general environment. This includes agents, markers, parts, tools, assembly sequences, visualizations, conditions, and events. Through the high degree of familiarity and interoperability, the web interface is easy to use

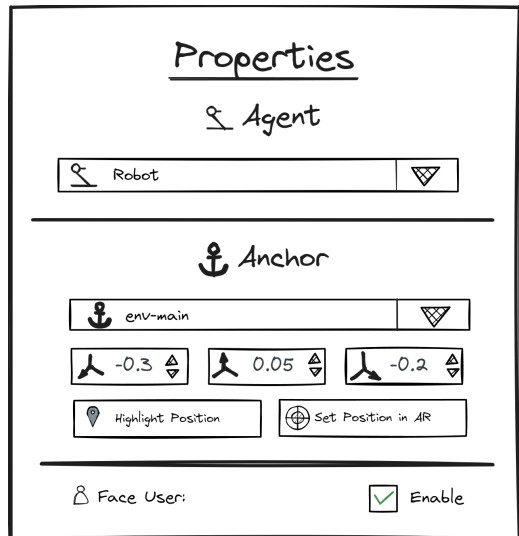

**Figure 2: A sketch of properties for a robot state visualization. An example of the visualization can be seen on Figure 1 (center and right picture). Three properties can be edited for this visualization:** *agent, anchor,* **and** *face user. Agent* **determines which robot the status is displayed from (relevant in a multi-robot scenario). The** *anchor* **is used to position the visualization relative to a known position (e.g. a QR-code or part of the robot). Finally,** *face user* **can be checked if the panel automatically should face the user.**

and can allows importing of existing data (e.g., parts, tools, assembly sequences) from other systems. Once imported, users can then add additional information, such as the physical location of the specific tool by switching from the web interface to the AR interface. Figure 4 shows a sketch of what the interface for authoring of visualizations, conditions, and events will look like.

*AR interface.* The AR interface is responsible for presenting the user with all active visualizations. Hand and gaze interaction is supported for authoring tasks that require spatial manipulations, for example when defining a 3D anchor point in the real world. As mid-air interaction is generally not well-suited for 2D menus (e.g., to improve ergonomics), we aim to minimize the need for traditional interface elements in AR.

### 3.3 Authoring Phases

We envision that the system will be used in three phases: *initial setup, authoring,* and *execution.* We expect that each phase will utilize the web and AR interfaces to a different degree (see Figure 3). During *initial setup,* information is fed into the system from other sources (e.g., PLM-systems) and static information is provided, such as the type of robot(s) and their position(s). The initial setup is likely done at the desktop interface.

The user can then transitions to the workspace with the robot for the *authoring* phase, where the design of the final user experience is created. We imagine that this will be done in an iterative manner, where visualizations and other elements are gradually added and

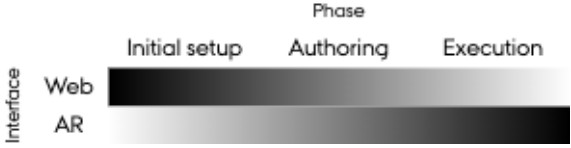

**Figure 3: Expected utilization of each interface in the authoring phases, with black indicating a high expected utilization.**

adjusted to fit the context of the task. Users can therefore benefit equally from the web interface for adjusting properties using the tablet, and the AR interface for adjusting elements with regard to their real-world context. To better support this switch, we proposed to mount the tablet on a fixture in proximity to the robot, allowing for a suitable perspective of the workspace and keeping the user's hands free.

Lastly, in the *execution* phase, an operator performs the task in collaboration with the robot, thereby perceiving the previously authored AR content. During this phase the operator might make ad-hoc changes, e.g., due to a spontaneous change in procedure, or personal preferences. However, here authoring abilities are intended to be limited.

It should be noted that these phases do not necessarily need to be completed by the same user.

## 4 DISCUSSION AND OPEN CHALLENGES

While our vision provides a general outlook to author AR-based HRC workflows, there are many opportunities and challenges left that can be addressed in future works regarding AR guidance, HRC authoring, and hybrid user interfaces.

*Clutter & Occlusion.* As prior work [1, 22] has demonstrated, using AR for guidance in HRC has many benefits, such as showing a 3D preview of where objects have to be placed, or visualizing anchors and targets within the user's workspace. However, a user's workspace can easily become cluttered with digital objects, which may be especially problematic during authoring, where the user has an overview over all necessary steps. Our vision addresses this by grouping steps into *conditions,* but other approaches may be more useful or user-friendly, such as proximity-based heuristics or focus+context visualization techniques. Similarly, showing a 3D preview for object placement can be beneficial, but also occludes the real-world object [22]. Here, a continuous adaption of the virtual preview (e.g., by fading out the object or transforming the full object into a wireframe as the real object gets closer) may be promising.

*Improving Author Guidance.* In-situ authoring in AR offers many opportunities for providing heuristic- or artificial-intelligence-based suggestions. For example, Belo et al. [2] demonstrated the utility of visualizing the ergonomic costs of interaction in AR. Ergonomics have been considered in the context of HRC [1], but could be factored into the authoring process (e.g., by visualizing the most viable ergonomic position of the robot arm in AR during authoring). Alternatively, the authoring system could detect and visualize inefficient or dangerous procedures and offer alternatives, thereby improving the authored procedure. Real-world evaluation of the proposed

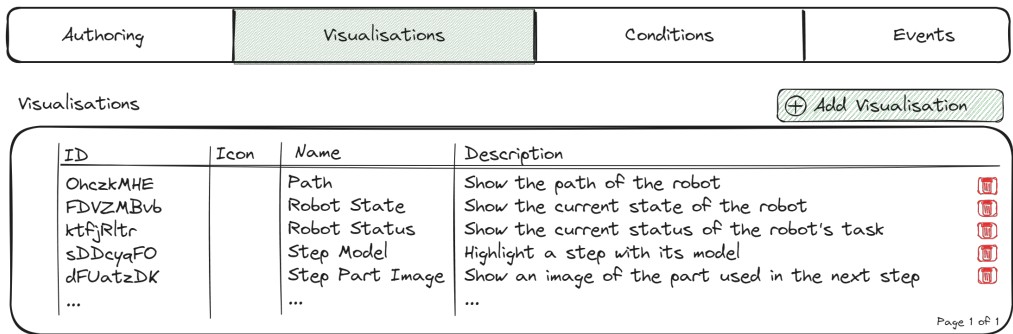

**Figure 4: A sketch of a list of created visualizations.**

approaches is needed, to validate their suitability for addressing training requirements and identify practical challenges.

*Coordination of Interface Use.* AR opens up a large design space for interaction, for example by allowing users to place controls anywhere in their workspace or attach a menu to their wrist. Yet, these interactions can be problematic, as they can be easily triggered inadvertently, especially when the user is handling real world objects during an assembly process. In contrast, the tablet in our system may be more limited in terms of placement, but provides a very explicit input modality that may be more suitable in HRC scenarios. Further studies are needed to investigate how each input and output modality can be used to their full potential.

Similarly, further research is necessary to investigate the learning curve of using our system and the cost of switching between interfaces. For example, we should help users better transition between these devices (cf. [15]), establish semantic connections, and fulfill user expectations: For example, configuring the properties of a 3D object on the web interface might force users to constantly switch their visual attention between the interface and actual 3D object in the AR environment. In this context, off-screen visualizations or visual linking techniques may be beneficial, but could lead to more clutter within the user's AR workspace.

*Beyond Assembly With Robot Arms.* While our vision initially focuses on the authoring experience for collaborative assembly with robot arms, we see potential in applying the same system to other areas. For example, support for mobile robots could be added, provided that suitable tracking is supported. Although fiducial markers (e.g., QR codes) already offer decent stability and precision for this scenario, current AR hardware still has significant limitations (e.g., low frequency of marker detection) that make a real-world evaluation difficult. One could also imagine using the authoring tool to supervise robot programming, though that will likely require new types of visualizations and data integration. Finally, we can imagine a setup where the operator is supervising multiple robots over a larger area, compared to the single workstation described in this paper. While parts of our system would then need to be altered for this different setup, the concept of visualizations, events, and conditions should remain applicable, e.g., by conveniently providing views with level of detail based on the operator's distance to the workstation (overview when distant, detailed view when close). Supporting a variety of robots may also permit exploring a range of other application scenarios where AR information displays can aid control and collaboration, such as in healthcare and medicine, outdoor exploration and monitoring, construction, or art displays.

## 5 CONCLUSION

We propose a system for in-situ authoring of augmented reality guidance for human-robot collaboration through a hybrid user interface. We thereby leverage the familiarity and advanced capabilities of a desktop computer or tablet for menu interaction, and an AR headset for 3D visualization and spatial configuration of virtual content through mid-air interaction. Our system addresses limitations from previous systems by (1) extending the possible AR guidance to include assembly instructions; (2) establishing two-way communication between robot and operator (e.g., by defining actions for the operator); and (3) combining strengths of multiple types of devices and user interfaces to provide a better user experience. With this paper we aim to highlight some of the most central opportunities and challenges of hybrid user interfaces for authoring of AR-supported HRC systems. We intend to realize our system in future work and use the functional prototype to investigate its usefulness in the real world. Our system shall thereby provide a tool for easy creation of different HRC setups and various AR feedback types, for extended future analysis and evaluation with users.

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
