# OpenReview forum: "Proposing a Hybrid Authoring Interface for AR-Supported Human-Robot Collaboration"
_humanrobotinteraction.org/HRI/2024/Workshop/VAM-HRI — VAM-HRI 2024 Oral_

### Official Review · Reviewer_V6pV · 2024-02-20
**Accept**

**Rating:** 9
**Confidence:** 5

**Review:**

This submission introduces a novel system that leverages augmented reality (AR) to facilitate human-robot collaboration (HRC) through a hybrid user interface. This system combines the strengths of desktop computers, tablets, and AR headsets to enable in-situ authoring of AR content for guiding assembly tasks in manufacturing environments. It aims to improve the efficiency and intuitiveness of HRC by allowing users to directly interact with and program robots in a shared workspace, enhancing communication and collaboration between humans and robots.

## Strengths:
- **Innovative Approach:** The paper presents a pioneering method to integrate AR into HRC, enhancing user experience and task efficiency.
- **Comprehensive System Design:** It covers a broad range of interaction modalities, providing flexibility in how users can author AR content.
- **Potential for Wide Application:** The system's design has implications for various industrial and research applications, promoting more intuitive and effective human-robot interactions.

## Weaknesses:
- **Implementation and Evaluation Details:** The paper might benefit from more detailed information on the implementation and comprehensive evaluation results to validate the proposed benefits.
- **User Learning Curve:** The complexity of the hybrid interface may introduce a learning curve that could affect the initial adoption rate among non-technical users.

## Recommendations for Improvement:
- **User Studies:** Conduct extensive user studies to assess the system's usability, effectiveness, and learning curve.
- **Expand Application Scenarios:** Explore and demonstrate the system's utility in diverse HRC scenarios beyond manufacturing to underline its versatility.
- **Enhance User Interface Design:** Simplify the interface to minimize the learning curve and improve accessibility for users with varying technical backgrounds.

In summary, I think this paper is a great fit for VAM-HRI, and I recommend acceptance.

---

### Official Review · Reviewer_3K2N · 2024-02-24
**Accept**

**Rating:** 8
**Confidence:** 4

**Review:**

The paper proposes a hybrid authoring interface based on their RoboVisAR project, for creating augmented reality (AR) content in human-robot collaboration (HRC) scenarios. It aims to bridge the gap between traditional desktop-based content creation and in-situ adjustments using an AR head-mounted display. This approach enables immediate updates and testing of AR content, facilitating quicker design iterations and potentially lowering the barrier for non-technical users to create AR-supported systems using a mix of HMD, tablet, and desktop systems.


Strengths:

1. Innovative hybrid interface that combines the strengths of desktop and AR environments for efficient AR content creation and adjustment.
2. Direct in-situ testing and adjustment of AR content can significantly speed up the design process and improve the relevance and accuracy of AR cues in real-world scenarios.
3. The approach is user-friendly, potentially making AR content creation accessible to a broader range of users, including those without extensive programming skills.

Areas of Improvement:

1. The reliance on multiple devices (desktop, tablet, AR headset) might complicate the setup and require users to switch between devices frequently, which could disrupt the workflow.
2. Real-world testing and validation of the proposed system are necessary to understand its practical challenges, training requirements (mostly during initial setup and authoring), and limitations in various HRC contexts.

In summary, I think this paper is a good fit for VAM-HRI, and I recommend acceptance.

---

### Decision · Program_Chairs · 2024-02-26

Accept (Oral)